# TASK-ORIENTED MULTI-VIEW REPRESENTATION LEARNING

## ABSTRACT

Multi-view representation learning aims to learn a high-quality unified representation for an entity from its multiple observable views to facilitate the performance of downstream tasks. A fusion-based multi-view representation learning framework consists of four main components: View-specific encoding, Single-view learning (SVL), Multi-view learning (MVL), and Fusion. Recent studies achieve promising performance by carefully designing SVL and MVL constraints, but almost all of them ignore the basic fact that *effective representations are different for different tasks, even for the same entity*. To bridge this gap, this work proposes a **T**ask-**O**riented **M**ulti-**V**iew **R**epresentation **L**earning (TOMRL) method, where the key idea is to modulate features in the View-specific encoding and Fusion modules according to the task guidance. To this end, we first design a gradient-based embedding strategy to flexibly represent multi-view tasks. After that, a meta-learner is trained to map the task embedding into a set of view-specific parameters and a view-shared parameter for modulation in the Encoding and Fusion modules, respectively. This whole process is formalized as a nested optimization problem and ultimately solved by a bi-level optimization scheme. Extensive experiments on four multi-view datasets validate that our TOMRL consistently improves the performance of most existing multi-view representation learning approaches.

## 1 INTRODUCTION

Learning a high-quality representation for target entities is a fundamental and critical problem that facilitates a variety of downstream tasks, such as classification tasks Jia et al. (2021); Han et al. (2023b), clustering tasks Zhong & Pun (2022); Lin et al. (2021), retrieval tasks Chen et al. (2017; 2023), and so on. However, in the real world, entities are generally so complex that we cannot model them directly and efficiently, but merely obtain observations from partial perspectives with the assistance of specific sensors Li et al. (2019a). For example, a bird can be represented by a piece of audio containing a call, a video of a flight, or a few candid photos, and similarly, it can be identified by shape, color, or texture. These multi-modal or multi-feature data derived from different sensors are collectively known as multi-view data, which each contain partial information and are combined as a representation of the bird entity. Multi-view representation learning is one such science that aims to investigate how to learn a unified entity representation from multi-view data and leverage it for downstream tasks Zhao et al. (2017); Huang et al. (2021); Yan et al. (2021).

As a classical topic, numerous excellent multi-view representations have been proposed and widely used. Early correlation-based approaches, such as CCA Chaudhuri et al. (2009) and its variants, align two views by constraining correlations Houthuys et al. (2018); Guo & Wu (2019); Uurtio et al. (2019). DCCA Andrew et al. (2013) extends it and lifts the limitation on the number of views by utilizing deep networks. Such an alignment-based approach inherently preserves consistent information across multiple views. In addition to the consistency, several studies have advocated that complementary information contained in different views is also beneficial for learning entity representations. For example, the CSMSC Luo et al. (2018) and DSS-MSCZhou et al. (2020) extract and fuse the features of these two components as the final entity representation, respectively. Further, recent approaches propose the notion of redundancy to remove information that is irrelevant or even harmful to the learning of a unified representation, such as noise, background, etc. MFLVC Xu et al. (2022) and MetaViewer Wang et al. (2023) separate and filter out the redundant information through a multi-level network architecture and bi-level optimization process, respectively. E²MVSC

explicitly decouples the information of consistency, complementarity, and redundancy using a two-head network with self-supervised constraints and filters out redundant information while retaining the other two via the information bottleneck principle. Existing solutions have made satisfactory advances in "how to learn a unified representation from multi-view data", but almost all of them neglect "the proper utilization of these representations in downstream tasks".

From the perspective of handling tasks, the above multi-view methods fall into two categories. One is learning the unified representation for a set of multi-view data at once, which is usually used for clustering tasks Zhang et al. (2019); Zheng et al. (2022); Lin et al. (2023). The advantage is that it potentially learns information about the data distribution (the current task), while the disadvantage is that it is not flexible enough to deal with out-of-sample data, which makes it almost unusable in real-world scenarios Busch et al. (2020); Zhang et al. (2021). The other one follows a more general form that focuses only on learning the corresponding uniform representation for entities, which can easily handle out-of-sample data and is therefore widely used for tasks such as clustering, classification, and retrieval Xu et al. (2022); Wang et al. (2023). The drawback is also obvious: valid representations should be different in different tasks. For example, a green cat should obviously be classified into different clusters in a "cat-dog" clustering task and a "color" clustering task. In other words, two entities that are grouped together for some tasks may be far apart for others.

In order to combine the advantages of the above two types of learning paradigms, this work provides a meta-learning-based solution and learns task-oriented multi-view representations, which can both flexibly handle out-of-sample data and help multi-view features (or unified representations) be better used for solving downstream tasks. Specifically, we first define an unsupervised multi-view task in an *episode* fashion and provide an embedding strategy for the multi-view task using the Fisher Information Matrix (FIM) Achille et al. (2019). Afterwards, a meta-learning model is built that receives task embeddings and maps them into two sets of *shift* and *bias* parameters for modulating view-specific features and unified entity representation, respectively. The meta-learning and multi-view learning processes are modeled as a nested optimization problem and ultimately solved by a bi-level optimization scheme. Note that this work looks at making the multi-view representations better adapted to downstream tasks, rather than how to extract unified representations from multi-view data (which has been well-studied in existing work). Thus, TOMRL can be easily integrated into existing multi-view methods to improve the quality of the unified representation. Experiments on both few-shot and routine tasks demonstrate the effectiveness of our TOMRL. The key contributions are

- We propose a Task-Oriented Multi-view Representation Learning method (TOMRL) from a meta-learning perspective. To the best of our knowledge, this could be the first exploration of "how representations from multiple views can better serve the task".

- TOMRL defines and embeds an unsupervised multi-view task in an *episode* fashion and designs a meta-learner for modulating the view-specific features and unified entity representations with the task guidance.

- TOMRL models meta-learning and multi-view learning as a nested bi-level optimization, where the high-level meta-learns the shift and bias parameters at task-level and low-level modulates and improves the multi-view representation based on them.

- TOMRL is compatible with most existing multi-view representation learning methods. Experimental results show that our method consistently improves the performance of downstream tasks for both few-shot and routine tasks.

## 2 A BASIC FUSION-BASED MULTI-VIEW LEARNING PIPELINE

Given an entity $x = \{x^v\}_{v=1}^V$ with $V$ views, sampled from the data distribution $p(x)$, where $x^v \in \mathbb{R}^{d_v}$ denotes the $v$-th view of the entity $x$, and $d_v$ is its dimension. Multi-view representation learning aims to learn a mapping from multiple views to a unified entity representation, $f : x \to H \in \mathbb{R}^{d_H}$, where $d_H$ is the dimension of the unified representation $H$. A high-quality unified representation integrates useful information from multiple views and represents entities in a more comprehensive way than any of them, and thus can be used in a variety of downstream tasks. Here we take the example of classification and clustering tasks, which are fundamental tasks in supervised and unsupervised learning settings, respectively. For the classification task, the entities usually correspond to manually labeled category labels $y$, and the learning goal transforms to pre-

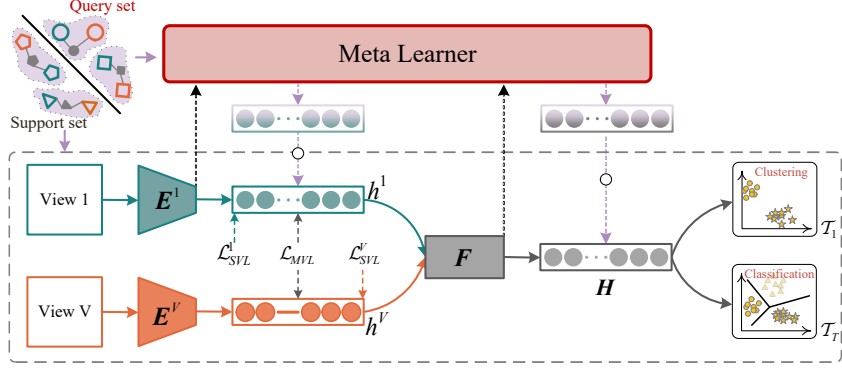

Figure 1: The overall framework of our TOMRL.

dicting the correct labels $y$ for $x$ based on $H$. For clustering tasks without any pre-labeled supervised information, the goal is to assign entities to different clusters where entities within the same cluster share some of the same attributes and vice versa. Either way, a high-quality, discriminative, unified entity representation is essential for downstream tasks.

DeepMVC Trosten et al. (2023) presents a general framework for multi-view clustering, and we extend here slightly to fusion-based multi-view representation learning Li et al. (2019a) [1]. As shown by the dashed box in Fig. 1, this framework consists of five modules. Along the data flow, they are:

- *View-specific Encoding*, or *Encoding* for short, is equipped with a (deep neural network) encoder $E^v$ with the trainable parameter $\theta_{h^v}$ for each view, and their task is to produce the view-specific representation $h^v = E^v(x^v; \theta_{h^v})$ from raw multi-view data.

- *Single-view Learning (SVL)* component consists of a set of auxiliary tasks designed to help optimize view-specific encoders $\{E^v\}_{v=1}^{V}$ to better learn view-specific features. Each auxiliary task is specific to its designated view, and is isolated from all other views.

- *Multi-view Learning (MVL)* is similar to SVL in that they both aim to constrain encoder training. The difference is that the auxiliary tasks in MVL are required to utilize all views simultaneously, allowing the model to learn features across views.

- *Fusion* module combines view-specific representations into the unified entity representation $H$ shared by all views. Fusion is typically done using a (weighted) average Li et al. (2019b); Trosten et al. (2021), or by concatenation Huang et al. (2019b); Xin et al. (2021); Xu et al. (2021). More complex fusion strategies are also possible, such as attention mechanisms Zhou & Shen (2020), meta-learning Wang et al. (2023), etc.

- *Evaluation* module uses the learned entity representations for downstream tasks. For example, Evaluation can consist of traditional clustering methods in clustering tasks, such as k-means Xu et al. (2017) or Spectral Clustering Shi & Malik (2000). In addition to these "train-then-evaluate" two-stage pipelines, some Evaluations that are integrated and jointly trained with other modules are also permissible.

Following this framework, the learning of the mapping $f$ from multi-view data to a unified representation $H$ can be formalized as:

$$f := H = F(\{\underbrace{E^v(x^v; \theta_{h^v})}_{h^v}\}_{v=1}^{V}; \theta_H), s.t. \underset{\theta_{h1}, \ldots, \theta_{hV}, \theta_H}{\arg\min} (\mathcal{L}_{SVL}^{1} + \cdots + \mathcal{L}_{SVL}^{v} + \mathcal{L}_{MVL}) \quad (1)$$

where $\mathcal{L}_{SVL}^{v}$ and $\mathcal{L}_{MVL}$ denote the loss functions involved in auxiliary tasks in SVL and MVL modules, respectively. Following the investigation of previous works, existing studies are devoted

---

[1]This extension is reasonable. In the unsupervised setting, SVL and MVL degenerate into SV-SSL and MV-SSL mentioned in DeepMVC Trosten et al. (2023) follow the self-supervised learning (SSL) paradigm: the learner is trained by designing a series of pretest tasks. As for supervised settings, SVL and MVL are dominated by label prediction tasks, assisted by pre-test tasks Trosten et al. (2021); Zhou & Shen (2020).

to elaborate various auxiliary tasks as well as various instances of $\mathcal{L}_{SVL}^{v}$ and $\mathcal{L}_{MVL}$ in SVL and MVL modules. Representative ones are, for example, the information bottleneck constraints Wan et al. (2021); Federici et al. (2020), the contrastive constraints Lin et al. (2021; 2023) and the MCR$^2$ discriminative constraints Yu et al. (2020a). These undoubtedly enhance the capability to derive a unified representation from multiple views. However, little research has focused on how learned representations are better adapted to the task, as reviewed in the previous section.

## 3 TASK-ORIENTED MULTI-VIEW LEARNING

To fill this gap, this work proposes a TOMRL method and investigates how to learn task-oriented multi-view representations, which is achieved by modulating view-specific and unified representations in Encoding and Fusion modules, respectively. The basic principles followed are twofold, for different tasks, a) the effective components are different in each view and b) the fusion process of features may also be inconsistent. Therefore, more explicitly, our TOMRL focuses on the Encoding and Fusion modules, which not only do not conflict with existing work on SVL and MVL, but also further enhance their performance in downstream tasks. Formally, multi-view representation learning with TOMRL on a task $t$ can be rewritten from Eq. 1 as:

$$f_t := H_t = F(\{\underbrace{E^v(x^v; \theta_{h^v}, w_{h_t^v})}_{h_t^v}\}_{v=1}^{V}; \theta_H, w_{H_t}), s.t. \underset{\substack{\theta_{h^1}, \ldots, \theta_{h^V}, \theta_H, \\ w_{h_t^1}, \ldots, w_{h_t^V}, w_{H_t}}}{\arg\min} (\mathcal{L}_{SVL}^{1} + \cdots + \mathcal{L}_{SVL}^{v} + \mathcal{L}_{MVL}) \quad (2)$$

where $w_{h_t^v}$ and $w_{H_t}$ denote additional introduced modulation parameters, where the former are view-specific representations of modulation parameters following the first principle and the latter are uniform representations of modulation parameters following the second principle. Here, we use FiLM Perez et al. (2018) to instantiate these two modulation processes. Formally, the modulation parameter consists of two parts, the *Scale s* and the *Shift b*, i.e., $w_{h_t^v} := \{s_{h_t^v}, b_{h_t^v}\}$ and $w_{H_t} := \{s_{H_t}, b_{H_t}\}$. For the given target task t, the modulated $v$-$th$ view representation and the unified representation are respectively: $h_t^v = h^v \odot s_{h_t^v} + b_{h_t^v}$ and $H_t = H \odot s_{H_t} + b_{H_t}$. It is worth noting that any form of modulation is allowed, e.g. sigmoid (gate) function Vuorio et al. (2019) and attention-based (softmax) modulation Mnih et al. (2014); Vaswani et al. (2017). More experimental comparisons of modulation strategies can be found in Sec 4.

### 3.1 A META-LEARNER FOR MODULATING PARAMETERS

The above modulation parameters are expected to introduce task bias for multi-view representations and are therefore derived from a meta-learner $M$ in this work, i.e.,

$$\underbrace{\{s_{h_t^v}, b_{h_t^v}\}_{v=1}^{V}}_{w_{h_t^v}}, \underbrace{s_{H_t}, b_{H_t}}_{w_{H_t}} = M(\{z_t^v\}_{v=1}^{V}, Z_t; \Theta), \quad (3)$$

where $\Theta$ is the meta (learner) parameter, $\{z_t^v\}_{v=1}^{V}$ and $Z_t$ are the embedding of task $t$ from the perspective of $v$-$th$ view and entity, respectively. Meta-learning is a promising paradigm for learning meta-knowledge from numerous historical tasks to quickly tackle unseen new tasks. Typical meta-learning strategies consist of a meta-learner and a base learner trained with an *episodic* setting. *Episode* consists of tasks, which further contain support sets and query sets. The base learner learns task-specifically on the support set of each task, and the meta-learner learns task-level meta-knowledge by integrating the loss on query set of multiple tasks, where the meta-knowledge can be anything in the training, such as parameters, hyper-parameters, or even data. Along this idea, we first extend the task definition in the previous work Huang et al. (2019a) and define two multi-view tasks with *episode* manner, and later encode the tasks with the help of a gradient-based strategy and design a new auxiliary task for the meta-learner. After that, we embed the multi-view tasks based on a gradient strategy and design a new auxiliary task for meta-learner training.

#### 3.1.1 MULTI-VIEW TASK WITH EPISODE

To emphasize the universality of the proposed TOMRL, we consider both supervised and unsupervised perspectives and define multi-view classification and multi-view clustering tasks with *episode* as illustrative instances.

- **Supervised Multi-view Classification Task** $T_{cls} = (N, K, K', S, Q, V)$ comes with two disjoint sets of data. Labeled **support** set $S_{T_{cls}} = \{(x_i^{s1}, \ldots, x_i^{sV}, y_i^s)\}_{i=1}^{NK}$ and Labeled **query** set $Q_{T_{cls}} = \{(x_i^{q1}, \ldots, x_i^{qV}, y_i^q)\}_{i=1}^{NK'}$ containing same $N$ classes, but disjoint $K$ and $K\prime$ samples, respectively. For such tasks, the base learner trains on the support set and makes predictions $\hat{y}_i^q$ on the support set, and the meta-learner updates by integrating the query errors across tasks.

- **Unsupervised Multi-view Clustering Task** $T_{clu} = (N, K, S, Q, V, V')$ consists of $N$ classes and $K$ unlabelled samples pre class, which are further constructed as two disjoint data sets from the view perspective. Unlabeled **support** set $S_{T_{clu}} = \{(x_1^v, \ldots, x_M^v)\}_{v=1}^{V'}$ and Unlabeled **query** set $Q_{T_{clu}} = \{(x_1^v, \ldots, x_M^v)\}_{v=V'}^{V}$. For such tasks, the base learner clusters the support set into $N$ clusters and maps the query samples to one of the predicted clusters, and the meta-learner updates by integrating the consistency error [2] on the query set across tasks.

### 3.1.2 MULTI-VIEW TASK EMBEDDING

Note that when $N$ and $K$ are small, the above task degenerates into a typical setting for few-shot learning, also known as the N-way K-shot task. In this work, there is no restriction on the number of $N$ and $K$, so the next difficulty is how to learn flexibly from multi-view tasks with different scales. To solve it, we take inspiration from Achille et al. (2019); Wang et al. (2021) and produce scale-consistent task embeddings based on the gradient information defined by the Fisher Information Matrix (FIM) of the Encoding and Fusion parameters. Taking a classification task $t$ as an example, the task embedding from the $v$-$th$ view perspective is defined as the FIM of the $v$-$th$ encoder $E^v$, which can be computed as

$$z_t^v := FIM_{\theta_{h^v}} = \mathbb{E}_{x^v, y \sim \hat{p}_t(x) p_t^{E^v}(y|x^v)}[\nabla_{\theta_{h^v}} \log p^{E^v}(y|x^v) \nabla_{\theta_{h^v}} \log p^{E^v}(y|x^v)^T] \quad (4)$$

where $\hat{p}_t(x)$ is the empirical distribution defined by the support set $S_t$. FIM actually measures the response of the parameters to the task utilizing the gradient Achille et al. (2019), so the classification error described above can be viewed as a specific instance of $\mathcal{L}_{SVL}^v$, meaning it can be easily computed in the multi-view framework with the $\mathcal{L}_{SVL}^v$. Similarly, the task embedding $Z_t$ from entity perspective can be computed as the FIM of the Fusion module $F$ with the $\mathcal{L}_{MVL}$.

### 3.1.3 A NEW VIEW INDEX PREDICTION AUXILIARY TASK

So far, the meta-learner can receive gradient-based task embeddings and derive modulation parameters for both the Encoding and Fusion modules. The last potential risk is to generate modulation parameters for different views with one meta-learner, which may confuse the knowledge at view-level. To this end, a simple view index recognition auxiliary task was additionally used. In addition to the modulation parameters, the meta-learning predicts the corresponding view index for the input task embedding with a recognition head. The view discriminability of meta-learning is increased by minimizing the recognition error, i.e., $\mathcal{L}_{IND} = \sum_{v=1}^{V} -vlog(v|h_t^v)$ Thus, the total loss function for meta learner is $\mathcal{L}_{meta} = \mathcal{L}_{base} + \mathcal{L}_{IND}$, where $\mathcal{L}_{base} = \mathcal{L}_{SVL}^1 + \cdots + \mathcal{L}_{SVL}^v + \mathcal{L}_{MVL}$ is abbreviated from Eq. 2.

### 3.2 BI-LEVEL OPTIMIZATION FOR META-LEARNING AND MULTI-VIEW LEARNING

Following previous works Finn et al. (2017); Shu et al. (2019), we train the parameter of meta learner and fine-tune the pre-trained parameter of multi-view framework in a bi-level optimization fashion, which can be formalized as

$$\Theta^* = \arg\min_{\Theta} \mathcal{L}_{meta}(\underbrace{\arg\min_{\theta} \mathcal{L}_{base}(\theta; \Theta)}_{\theta^*(\Theta)}), \quad (5)$$

---

[2]This is essentially a self-supervised constraint, where the meta-learner is trained to learn the task information using the self-supervised information that "the clustering results are consistent across different views of the entity".

---

**Algorithm 1** The framework of our TOMRL.

---

**Require**: Pre-trained DeepMRL parameters $\theta_{h^1}, \ldots, \theta_{h^V}$ and $\theta_H$, meta parameters $\Theta$, Dataset $D$, the number of views $V$, the number of task per episode $T$, maximum iteration steps in outer-level optimisation $O$, maximum iteration steps in inner-level optimisation $L$.

1: Initialize $\Theta$;
2: **for** $o = 1, \ldots, O$ **do**
3:     *# Outer-level optimization*
4:     Sample batch of tasks $\{\mathcal{T}_t = (S_t, Q_t)\}_{t=1}^T \sim p(\mathcal{T})$ constructed from $D$.
5:     **for** $t = 1, \ldots, T$ **do**
6:         Obtain task embedding $z_t$ via Eq. (4)
7:         Obtain modulation parameters $w_{h_t^v} = \{s_{h_t^v}, b_{h_t^v}\}$ and $w_{H_t} = \{s_{H_t}, b_{H_t}\}$ via Eq. (3).
8:         **for** $l = 1, \ldots, L$ **do**
9:             *# Inner-level optimization*
10:            **for** $v = 1, \ldots, V$ **do**
11:               Obtain view-specific representation $h^v = E^v(x_i^{sv}, \theta_{h^v})$.
12:               Modulate view-specific representation $h_t^v = h_t \odot s_{h_t^v} + b_{h_t^v}$.
13:            **end for**
14:            Obtain the unified representation $H = F(h^v, \Theta_H)$.
15:            Modulate the unified representation $H_t = H \odot s_{H_t} + b_{H_t}$.
16:            Update $\theta_{h^1}, \ldots, \theta_{h^V}$ and $\theta_H$ via Eq. (7).
17:         **end for**
18:     **end for**
19:     Update $\Theta$ via Eq. (6).
20: **end for**
21: Return the optimal parameters $\theta_{h^1}, \ldots, \theta_{h^V}$ and $\Theta$.

---

Overall, the outer-level optimization expects to learn an optimal set of meta-learner parameters, which produces the modulation parameters mentioned in Eq. 2 for modulating the multi-view representation. The inner-level optimization can handle specific tasks better with the help of these modulations. Since the meta-learner learns task-level knowledge across multiple tasks, the above process is known as task-oriented multi-view representation learning. More specifically, in the $o\text{-}th$ outer-level optimization, the parameters of the meta-learner are updated as

$$\Theta^o = \Theta^{o-1} - \frac{\beta}{T} \sum_{t=1}^T \nabla_\Theta \mathcal{L}_{meta}\left(Q_t;\ \theta_t(\Theta^{o-1})\right), \tag{6}$$

with the updated meta parameters after the $o\text{-}th$ outer-level optimization, the inner-level optimization modulates the multi-view representation to adapt (the support set of) on each task. For task t, the update of the base parameters in the $l\text{-}th$ inner-level optimization is

$$\theta_t^l = \theta_t^{l-1} - \alpha \nabla_\theta \mathcal{L}_{base}(S_t;\ \theta_t^{l-1} \circ \Theta^o). \tag{7}$$

The $\alpha$ and $\beta$ are the learning rates of the base and meta-learning, respectively. The overall flow of our TOMRL is presented in Alg. 1.

## 4 RELATED WORKS

This work focuses on using feature modulation strategies to address multi-view representation learning in a meta-learning paradigm. The introductory chapter reviews several representative approaches to multi-view representation learning and clarifies the relationship with the work in this paper: the former focuses on modeling the representation, while the latter is concerned with the adaptation of the representation to the task. More multi-view learning approaches have been well summarized and discussed in recent surveys Yan et al. (2021); Trosten et al. (2023). In addition, we discuss in detail the differences and connections between our TOMRL and existing learning paradigms involving

multiple views and multiple tasks in Appendix A.1. In the following, we mainly review multi-view meta-learning and parameter modulation strategies.

**Meta learning vs. Multi-view Learning** These two paradigms focus on learning shared knowledge from multiple tasks and multiple views, respectively, so their combination is bi-directional. Multi-view meta-learning, or multi-modal meta-learning, is dedicated to exploiting the rich information contained in multiple views to improve the performance of the meta-learner in a few-shot or self-supervised scenario Vuorio et al. (2019); Ma et al. (2022). For example, a series of studies improve the robustness of prototype features in metric-based meta-learning methods with the help of additionally introduced modal information Xing et al. (2019); Pahde et al. (2021); Zhang et al. (2022). MVDG Yu et al. (2020b) designs a multi-view constraint to improve the generalization of gradient-based meta-learning, where multiple optimization trajectories learning from multiple views are employed to produce a suitable optimization direction for model updating. In contrast, some studies utilize meta-learning paradigms to facilitate multi-view representation learning. For example, SMIL Ma et al. (2021) uses Bayesian meta-learning to address the challenge of severely missing modalities. MetaViewer Wang et al. (2023) meta-learning a data-dependent multi-view fusion rule to model entity-view relationships. Our TOMRL belongs to the latter. Unlike MetaViewer, which meta-learns representations directly, TOMRL meta-learns a set of modal parameters that can be integrated into most existing multi-view methods and improve their task adaptability.

**Parameter modulation** Parametric modulation is a common strategy to flexibly adapt models to a variety of complex scenarios Feng et al. (2021); Brockschmidt (2020); Park et al. (2019). FiLM Perez et al. (2018) is a typical modulation strategy that scales and shifts the activated neurons by adding a multi-layer perceptron (MLP) for each layer. Based on this, MTL Sun et al. (2019) learns transferable parameters in order to quickly adapt to new tasks. MMAML Vuorio et al. (2019) modulates meta-learned initialization parameters to handle multi-modal tasks. Inspired by them, this paper uses FiLM to modulate multi-view representation. The difference is that these parameters are meta-learned across tasks and have task-oriented properties.

## 5 EXPERIMENTS

In this section, we present extensive experimental results to validate the effectiveness and flexibility of our TOMRL. The remainder of the experiments are organized as follows: Subsection 5.1 lists datasets, compared methods, and implementation details. Subsection 5.2 Subsection 5.3 investigates the performance of TOMRL in regular multiview tasks and cross-domain multiview tasks, respectively. Ablation studies and in-depth analyses are included in Subsection 5.4.

### 5.1 EXPERIMENTAL SETUP

**Datasets**. Four widely used datasets are adopted for our experiments, including Caltech 101-7, COIL-20, NoisyFashion, and EdgeFashion. Concretely, Caltech 101-7 Cai et al. (2013) is a subset of Caltech101 Fei-Fei et al. (2004), contains 7 classes and 6 handcrafted features as views. COIL-20 Nene et al. (1996) is a three-view dataset consisting of 1440 grayscale images belonging to 20 categories. NoisyFashion is generated using FashionMNIST Xiao et al. (2017), which contains two views: the raw image and its Gaussian noised image ($\sigma = 0.2$, the setting following Trosten et al. (2023)). EdgeFashion is another version of FashionMNIST. We employ the raw image and the edge-detected image of the same instance as two views.

**Baselines**. Along with our proposed TOMRL, three state-of-the-art multi-view clustering baselines from the last five years are selected for comparison, of which: (i) Deep Multimodal Subspace Clustering (DMSC) Abavisani & Patel (2018); (ii) Multi-view Spectral Clustering Network (MvSCN) Huang et al. (2019b); (iii) Simple Multi-View Clustering (SiMVC) Trosten et al. (2021). In addition to this, we compare the AE-KM, AECoDDC, and InfoDDC proposed by Trosten et al. (2023), as well as the results with TOMRL. These three approaches represent a simple baseline without constraints, with comparison constraints, and with information theoretic constraints, respectively, demonstrating the flexibility of TOMRL. To make the results more reliable, we report the average of 100 tasks in clustering and classification, respectively.

**Implementation details**. The implementation of TOMRL is based on the DeepMVC open source framework Trosten et al. (2023). All baselines adopt the recommended training hyper-parameters.

Table 1: Clustering results 5-way 5-shot multi-view tasks. Bold denotes the results of our TOMRL.

| Methods | NoisyFashion | | EdgeFashion | | COIL-20 | | Caltech7 | |
|---|---|---|---|---|---|---|---|---|
| | $ACC_{clu}$ | NMI | $ACC_{clu}$ | NMI | $ACC_{clu}$ | NMI | $ACC_{clu}$ | NMI |
| DMSC | 58.74 | 63.28 | 51.26 | 54.50 | 34.57 | 67.39 | 45.11 | 46.05 |
| MvSCN | 70.10 | 72.85 | 63.52 | 66.39 | 59.23 | 78.74 | 55.87 | 56.41 |
| SiMVC | 73.72 | 78.33 | 67.28 | 70.41 | 89.63 | 92.67 | 61.06 | 62.11 |
| AE-KM | 60.80 | 71.81 | 53.21 | 57.92 | 37.46 | 71.71 | 57.99 | 55.26 |
| **w/ TOMRL** | **62.31** | **74.14** | **55.73** | **59.06** | **40.25** | **73.33** | **59.85** | **57.77** |
| AECoDDC | 63.66 | 67.20 | 60.26 | 68.87 | 95.86 | 96.01 | 64.13 | 65.84 |
| **w/ TOMRL** | **66.25** | **70.06** | **62.38** | **71.28** | **96.10** | **98.41** | **65.10** | **66.54** |
| InfoDDC | 72.35 | 79.74 | 70.76 | 72.52 | 62.80 | 79.68 | 49.20 | 50.93 |
| **w/ TOMRL** | **75.80** | **82.93** | **72.35** | **77.89** | **65.75** | **80.26** | **51.25** | **51.38** |

Table 2: Classification results on 5-way 5-shot multi-view tasks. Bold denotes the results of our TOMRL.

| Methods | NoisyFashion | | EdgeFashion | | COIL-20 | | Caltech7 | |
|---|---|---|---|---|---|---|---|---|
| | $ACC_{cls}$ | Prec. | $ACC_{cls}$ | Prec. | $ACC_{cls}$ | Prec. | $ACC_{cls}$ | Prec. |
| DMSC | 72.84 | 70.58 | 63.66 | 58.03 | 77.38 | 72.99 | 65.28 | 61.43 |
| MvSCN | 88.32 | 82.36 | 70.82 | 62.87 | 80.71 | 78.25 | 63.57 | 62.70 |
| SiMVC | 95.86 | 88.27 | 78.58 | 66.29 | 98.90 | 98.26 | 71.84 | 69.09 |
| AE-KM | 64.51 | 77.13 | 59.97 | 60.91 | 78.42 | 71.71 | 62.13 | 63.33 |
| **w/ TOMRL** | **65.68** | **79.22** | **61.14** | **61.96** | **79.64** | **73.10** | **64.52** | **64.92** |
| AECoDDC | 82.84 | 74.05 | 81.87 | 66.07 | 97.25 | 95.73 | 65.64 | 67.64 |
| **w/ TOMRL** | **84.59** | **66.80** | **83.01** | **66.92** | **98.72** | **97.05** | **70.32** | **68.10** |
| InfoDDC | 94.84 | 89.33 | 76.22 | 68.39 | 87.76 | 83.23 | 66.33 | 57.74 |
| **w/ TOMRL** | **96.07** | **90.68** | **79.68** | **70.58** | **89.18** | **84.57** | **68.29** | **58.26** |

For the baseline with TOMRL, an additional meta-learning process is added after the regular training. Following the setup of previous works Finn et al. (2017); Antoniou et al. (2019), we sample five tasks per episode and set the learning rate to 0.001 and 0.0001 for the inner-level and outer-level optimization processes, respectively. The inner loop performs a single-step optimization, i.e., $L = 1$. Two downstream tasks, including clustering and classification, are employed in our experiments. Two popular clustering metrics, i.e., accuracy ($ACC_{clu}$) and normalized mutual information (NMI), are used to quantify the clustering effectiveness. For the classification task, two common metrics are used, including accuracy ($ACC_{cls}$), and precision (Prec.). Note that a higher value of these metrics indicates better clustering or classification performance.

## 5.2 RESULTS IN THE 5-WAY 5-SHOT MULTI-VIEW TASKS

To validate the benefits of TOMRL on cross-task learning, we first constructed regular 5-way 5-shot multi-view clustering and classification tasks. In this setting, the performance differences between the baseline methods w/ and w/o TOMRL both come from task bias. For the clustering task, we use the post-processing strategies recommended by the baseline method, such as AE-KM and AE-CoDDC using k-means and deep clustering algorithms, respectively Trosten et al. (2023), and the classification task is complemented by the SVM classifier Wang et al. (2023). Corresponding results are listed in Table 1 and Table 2. TOMRL brings consistent performance gains to the baseline approaches, suggesting that 1) TOMRL is independent of auxiliary tasks and fusion strategies; 2) TOMRL is robust to different task metrics and even to task types; and 3) TOMRL effectively learns and utilizes task bias information.

## 5.3 RESULTS IN CROSS-DOMAIN MULTI-VIEW TASKS

Further, we consider a more challenging cross-domain task setup where the training and testing tasks are sampled from different datasets. Specifically, two cross-domain scenarios are included, from NoisyFashion to EdgeFashion and from EdgeFashion to NoisyFashion, respectively. Table 3

Table 3: Clustering results of all methods. Bold and underline denote the best and second-best results, respectively.

| Methods | NoisyFashion → EdgeFashion | | | | EdgeFashion → NoisyFashion | | | |
|---|---|---|---|---|---|---|---|---|
| | $ACC_{clu}$ | NMI | $ACC_{cls}$ | Prec. | $ACC_{clu}$ | NMI | $ACC_{cls}$ | Prec. |
| AE-DDC | 48.26 | 43.86 | 50.81 | 49.25 | 44.50 | 50.82 | 46.12 | 58.03 |
| w/ TOMRL | 52.25 | 54.37 | 56.10 | 55.07 | 49.04 | 55.27 | 50.85 | 62.10 |
| AECoDDC | 48.58 | 51.22 | 62.33 | 47.58 | 45.15 | 48.98 | 62.78 | 55.26 |
| w/ TOMRL | 52.80 | **57.39** | **70.26** | 58.77 | 52.88 | 53.16 | 69.22 | 59.30 |
| InfoDDC | 53.74 | 56.95 | 58.35 | 56.20 | 51.39 | 54.25 | 68.58 | 62.17 |
| w/ TOMRL | **60.85** | 55.74 | 63.58 | **60.87** | **56.71** | **59.32** | **72.24** | **67.39** |

demonstrates the enhancements that TOMRL brings to the three baseline methods. Compared to the results in Table 1 and Table 2, the relative benefit of TOMRL is higher. One possible reason is that in cross-domain tasks, both clustering and classification tasks require more task-relevant information.

## 5.4 ABLATION ANALYSIS

**Ablation analysis.** We perform three ablation experiments on the COIL-20 dataset using AECoDDC as the baseline, including TOMRL without view-specific representation modulation (w/o $h_t^v$), without unified representation modulation (w/o $H_t$), and without view index prediction task ($\mathcal{L}_{IND}$). The results listed in Table 4 show that: 1) the self-supervised information provided by the auxiliary task slightly improves the results; 2) both modulation strategies are crucial for performance improvement.

Table 4: Ablation results on the loss function. Bold and underline denote the best and second-best results, respectively.

| Ablation | $ACC_{clu}$ | NMI | $ACC_{cls}$ | Prec. |
|---|---|---|---|---|
| Baseline | 95.86 | 96.01 | 97.25 | 95.73 |
| TOMRL w/o $h_t^v$ | 96.91 | 97.63 | 98.07 | 96.28 |
| TOMRL w/o $H_t$ | 96.57 | 97.28 | 98.66 | 96.35 |
| TOMRL w/o $\mathcal{L}_{IND}$ | 97.98 | 98.12 | 99.03 | 96.99 |
| TOMRL | **98.10** | **98.41** | **99.56** | **98.47** |

**Modulation mechanisms.** Tab. 5 compares several commonly used modulation strategies in the same data and task settings as in the previous subsection. Gate and Softmax are both prototype-dependent, non-linear, and non-parametric strategies. FiLM is substantially ahead of the other four strategies, validating the effectiveness of modeling the transformation from a unified entity to specific views with the affine transformation.

Table 5: Comparison of different modulation mechanisms. Bold and underline denote the best and second-best results, respectively.

| Modulation | $ACC_{clu}$ | NMI | $ACC_{cls}$ | Prec. |
|---|---|---|---|---|
| Baseline | 95.86 | 96.01 | 97.25 | 95.73 |
| Gate | 96.07 | 96.53 | 97.98 | 96.44 |
| Softmax | 98.18 | 98.02 | 98.58 | 97.26 |
| Scale | **98.92** | 98.39 | 99.13 | 97.90 |
| Shift | 97.58 | 97.87 | 98.21 | 96.82 |
| FiLM | 98.10 | **98.41** | **99.56** | **98.47** |

## 6 CONCLUSION

This work proposes a task-oriented multi-view representation learning method, TOMRL, to bootstrap existing methods to handle downstream tasks better through representation modulation. To this end, we resort to a gradient-based meta-learning paradigm and construct a meta-learner to generate modulation parameters. We further define multi-view tasks in an episodic manner, demonstrate a task embedding strategy based on gradients, and design a new view index prediction assistance task. Meta-learning and multi-view learning are ultimately formalized as a nested optimization problem and solved via a bi-level optimization paradigm. Extensive experiments validate that our TOMRL can be easily integrated into existing multi-view representation learning methods and bring consistent performance gains. Although this work is an initial attempt to introduce task bias in multi-view representation learning, promising results were obtained that may inspire subsequent multi-view representation learning approaches. Future work will focus on modeling view-task relationships and designing more comprehensive task encoding strategies.

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

# A  APPENDIX

## A.1  CONNECTIONS AND DIFFERENCES WITH MULTI-VIEW MULTI-TASK LEARNING (MVMTL) AND CONTINUOUS/LIFELONG MULTI-VIEW LEARNING (CMVL)

MVMTL Li & Huan (2018); Han et al. (2023a), CMVL Yu et al. (2023); Sun et al. (2018), and our TOMRL all consider both multi-view and multi-task scenarios, exploiting comprehensive feature representation of multiple views in each task as well as the task relationships of multiple related tasks. However, they are different in terms of learning objectives, key challenges, and practical settings:

- MVMTL follows the multi-task learning setting, where the learning objective is to improve performance on the main task through joint training on multiple related (auxiliary) tasks. Key challenges include selecting or designing auxiliary tasks, balancing loss functions across multiple tasks, etc. The counterpart is conventional (single-view) multi-task learning, with the difference that MVMTL utilizes comprehensive information (e.g., consistency and complementarity, etc.) from multiple views of each task. Once the multi-view information is well integrated, most MVMTL methods degrade to the conventional multi-task scenario Lu et al. (2020).

- CMVL follows the continuous learning setting, where the learning objective is to handle new tasks without performance degradation on previously learned tasks for a series of consecutive tasks. The key challenge is to overcome catastrophic forgetting. Similarly, the counterpart is conventional (single-view) continuous learning, with the difference that CMVL utilizes comprehensive information from multiple views of each task.

- Our TOMRL follows the meta-learning setting, where the learning objective is to rapidly handle new downstream tasks by learning meta-knowledge over multiple tasks. To this end, we construct task-level training and test sets on the multi-view dataset and train the model in an episodic fashion. The counterpart is conventional (single-view) meta-learning, with the difference that TOMRL considers both view-specific representations and fused unified representations in the meta-learning process. If the multi-view properties of the task are ignored, TOMRL degrades to a conventional meta-learning method Vuorio et al. (2019).

