# OpenReview forum: "Task-Oriented Multi-View Representation Learning"
_ICLR.cc/2024/Conference — Submitted to ICLR 2024_

### Official Review · Reviewer_fqyn · 2023-10-29

**Soundness:** 3 good
**Presentation:** 3 good
**Contribution:** 2 fair
**Rating:** 5
**Confidence:** 3

**Summary:**

This paper argues that the semantic features required for different tasks may vary when using the same representation. Therefore, it proposes a task-oriented multi-view representation learning method. The paper adopts a meta-learning paradigm, defines multi-view tasks, and retains both task-specific information for each view and unified information across views. Additionally, it introduces task bias for different tasks. This method can be integrated into existing multi-view representation learning methods and has shown performance improvements.

**Strengths:**

1. The motivation and idea presented in this paper are reasonable. The introduction of the paper effectively highlights that multi-view representations should focus on both the unified representation and the unique information of each view, and should adapt to different representations for different tasks.

2. This paper leverages a meta-learning paradigm to generate task bias and integrates it into existing multi-view representation learning methods, resulting in performance improvements.

**Weaknesses:**

1. The paper artificially defines downstream tasks for multi-view representation learning, but typically representation learning aims to acquire a general representation that can be fine-tuned for different specific tasks. Is it unfair to directly consider downstream tasks in the representation learning, and can the representations learned in this paper still perform well for new tasks that are not considered in the paper?

2. The paper is not very clear in explaining how the meta-learning paradigm generates task-specific biases and whether it can explain why using task bias generated by the meta-learning paradigm can improve the model's performance.

3. The experimental evaluations are not comprehensive enough in several aspects. The dataset used in this paper is not sufficiently diverse. For example, commonly used datasets in this field, such as NoisyMNIST, EdgeMNIST, Caltech20, and PatchedMNIST, were not tested. Additionally, the paper only integrates the method into three approaches, all from the same source paper. It is hoped that the authors can validate their method by integrating it into a wider range of methods to enhance its credibility.

**Questions:**

For major concerns including the problem/experimental setting, unclear description, and experimental evaluation, please see weaknesses for details.

There appears to be a minor error in the pseudocode. Should line 20 be placed after line 21?

---

> ### Author Response · Authors · 2023-11-23
> **Response to Reviewer fqyn**
>
> Thank you for your constructive comments.
>
> **Response to Weakness 1 (Experimental settings):**
>
> We apologize for some distress caused by the unclear experimental setup in the submitted manuscript. **It should be clarified that our experiments are fair because all methods (including our TOMRL) use the same data in the training and testing stages.** Specifically, both the comparison methods and baseline methods were allowed to train to convergence on the full dataset $D$. We then constructed $D$ as a set of (multi-view few-shot) tasks and randomly selected 100 tasks for testing. The remaining tasks were used for the training of the meta-learner in an episodic fashion, as shown in Alg. 1 in the manuscript. The well-trained TOMRL was also tested on the same 100 tasks. Note that these test tasks are never used to fine-tune any method to ensure a fair comparison. In other words, they are new and unseen tasks for both the compared methods and our TOMRL. This experimental setting ensures that the performance gains from TOMRL are derived from meta-learned task-level knowledge.
>
> **Response to Weakness 2 (Why TOMRL works):**
>
> As mentioned in the response to Weakness 1, we train the meta-learner additionally at the task level for the well-trained existing multi-view representation model. Indeed, the meta-learner is expected to learn a set of mappings from task embeddings to modulation parameters, in order to quickly adapt to unseen new tasks. More specifically, for each task in the meta-training stage, the meta-learner generates modulation parameters for the support set and performs clustering or classification on the query set using the modulated features. Minimizing the loss on the query set can induce the meta-learner to learn better mappings.
>
> **Response to Weaknesses 3 (Additional experimental results):**
>
> As a supplement, the following tables list the results of the TOMRL method on the four datasets suggested by the reviewer. Note that the results on the PatchedMNIST dataset are not included because PatchedMNIST is a subset of the MNIST dataset containing only the first three classes [1], which does not satisfy the settings of the 5-way task.
>
> Table: The ACC$_{clu}$ values on 100 5-way 5-shot clustering tasks
>
> |  | NoisyMNIST | EdgeMNIST | Caltech20 |
> | --- | --- | --- | --- |
> | AE-KM | 62.00 | 59.73 | 37.12 |
> | w/ TOMRL | 63.14 | 62.03 | 38.85 |
> | AECoDDC | 68.24 | 64.54 | 60.60 |
> | w/ TOMRL | 68.48 | 66.08 | 62.34 |
> | InfoDDC | 65.18 | 43.20 | 57.60 |
> | w/ TOMRL | 66.83 | 45.43 | 61.07 |
>
> Table: The NMI values on 100 5-way 5-shot clustering tasks
>
> |  | NoisyMNIST | EdgeMNIST | Caltech20 |
> | --- | --- | --- | --- |
> | AE-KM | 76.11 | 71.84 | 70.39 |
> | w/ TOMRL | 76.38 | 72.17 | 70.71 |
> | AECoDDC | 78.39 | 71.93 | 66.80 |
> | w/ TOMRL | 77.90 | 71.64 | 68.42 |
> | InfoDDC | 76.80 | 56.18 | 61.57 |
> | w/ TOMRL | 78.74 | 58.53 | 64.35 |
>
> [1] Trosten, Daniel J., et al. On the Effects of Self-supervision and Contrastive Alignment in Deep Multi-view Clustering. In *CVPR* 2023.
>
> **Response to Questions (Minor error):**
>
> Thanks for pointing out the error. We've fixed this error in the revised manuscript: switched lines 20 and 21 in pseudo-code.

---

### Official Review · Reviewer_d8EV · 2023-10-31

**Soundness:** 3 good
**Presentation:** 2 fair
**Contribution:** 1 poor
**Rating:** 3
**Confidence:** 5

**Summary:**

The authors propose a model to learn task-oriented multi-view representation. Based on the observation that almost all of current models ignore the basic fact that effective representations are different for different tasks, even for the same entity, they propose a Task-Oriented Multi-View Representation Learning (TOMRL) method.

**Strengths:**

Learning task-oriented multiview representation is important.

**Weaknesses:**

1.	The motivation is not clear. For example, there are many methods could learn task-oriented representation and multi-view representation, however, the authors only provide some examples which are not applicable for out-of-sample data or only learning the uniform representation for entities. It is not convincible.

2.	“A typical multi-view representation learning framework consists of four main components: View-specific encoding, Single-view learning (SVL), Multi-view learning (MVL), and Fusion.”  It is a very strong claim or assumption. The authors should provide a comprehensive study and moreover a formulation to unify these models is necessary.

3.	“how representations from multiple views can better serve the task” I think this is a very natural requirement in many models. So, I do not find any necessity or novelty for this claim.

4.	The writing and organization are not clear. It is difficult to understand the motivation and why the proposed model is good.

**Questions:**

The claim that there are four components is questionable, so the authors should provide more clear and strict evidence or analysis.

There are no theoretical results.

---

> ### Author Response · Authors · 2023-11-23
> **Response to Reviewer d8EV**
>
> Thank you for your constructive comments, especially on the multi-view representation learning framework, which is helpful to improve our work!
>
> **Response on Motivation and Innovation (Weaknesses 1 & 3 & 4):**
>
> To more clearly demonstrate our motivations and innovations, we review related research, especially in multi-view multi-task learning (MVMTL) and continuous multi-view learning (CMVL), which both involve multi-view and multi-task scenarios.
>
> - **MVMTL follows the multi-task learning setting**, where the learning objective is to improve performance on the main task through joint training on multiple related (auxiliary) tasks. The counterpart is conventional (single-view) multi-task learning, with the difference that MVMTL utilizes comprehensive information (e.g., consistency and complementarity, etc.) from multiple views of each task. Once the multi-view information is well integrated, most MVMTL methods degrade to the conventional multi-task scenario [1].
> - **CMVL follows the continuous learning setting**, where the learning objective is to handle new tasks without performance degradation on previously learned tasks for a series of consecutive tasks. Similarly, the counterpart is conventional (single-view) continuous learning, with the difference that CMVL utilizes comprehensive information from multiple views of each task.
> - Unlike the above two types of methods, our **TOMRL aims to learn the task-oriented multi-view representation** with the goal of learning task-level experience from numerous historical multi-view tasks, enabling the model to adapt quickly when dealing with unseen new multi-view tasks.
>
> **MOTIVATION:** Existing models for learning multi-view representations (especially in unsupervised settings) mostly focus on how to integrate comprehensive information from multiple views at the data level while neglecting fast adaptation on different tasks.
> **INNOVATION:** We provide a meta-learning perspective for task-oriented multi-view representation learning. 1) We define the ``episode'' form of supervised and unsupervised multi-view learning tasks, and construct meta-training and testing sets at the task level. 2) Based on the two basic principles mentioned in Section 3, we design a meta-learner for feature modulation and train it in an episodic fashion. 3) For an unseen new multi-view task, the well-trained meta-learner generates a set of modulation parameters that help the model quickly derive the uniform representation specific to this task.
>
> **Response on multi-view representation learning framework (Weakness 2):**
>
> The multi-view learning framework mentioned in this paper is extended by the deep multi-view clustering framework [2]. Thanks to your valuable comments, we rethink the rigor of this extension.
> Following the survey studies[3, 4], multi-view representation learning can be roughly divided into *alignment-based* and *fusion-based* methods. Let $x^1$ and $x^2$ denote the two views of sample $x$, alignment-based methods can be symbolized as $f(x^1;W_f)\leftrightarrow g(x^1;W_g)$, where $f$ and $g$ denote the feature extractors of view 1 and view 2 with corresponding parameters $W_f$ and $W_g$, respectively, and $\leftrightarrow$ denotes the alignment operation. Similarly, the fusion-based approach can be expressed as $H=F((f(x^1;W_f), g(x^1;W_g)); W_F)$, where $F$ is the fusion operation with the parameter $W_F$, and $H$ denotes the fused unfied representation. Although recent work [5] has attempted to integrate the two into a unified framework, it is clear that our TOMRL is strictly more suitable for fusion-based multi-view representation learning methods due to the fusion operations involved. Therefore, we modify ''the multi-view representation learning framework'' to ''the  fusion-based multi-view representation learning framework'' and clarify the research scope in the revised version.
>
> [1] Lu, Run-kun, et al. Multi-view representation learning in multi-task scene. *Neural Computing and Applications* 32 (2020): 10403-10422.
>
> [2] Trosten, Daniel J., et al. On the Effects of Self-supervision and Contrastive Alignment in Deep Multi-view Clustering. In *CVPR* 2023.
>
> [3] Li, Yingming, et al. A survey of multi-view representation learning. *IEEE TKDE* 31.10 (2018): 1863-1883.
>
> [4] Yan, Xiaoqiang, et al. Deep multi-view learning methods: A review. *Neurocomputing* 448 (2021): 106-129.
>
> [5] Wang, Ren, et al. MetaViewer: Towards A Unified Multi-View Representation. In *CVPR* 2023.

---

### Official Review · Reviewer_V7F6 · 2023-11-04

**Soundness:** 3 good
**Presentation:** 4 excellent
**Contribution:** 3 good
**Rating:** 6
**Confidence:** 4

**Summary:**

This paper presents a gradient-based embedding strategy to flexibly represent multi-view tasks. The authors propose a meta-learning-based solution and learns task-oriented multi-view representations, where meta-learning and multi-view learning are ultimately formalized as a nested optimization problem and solved via a bi-level optimization paradigm.

**Strengths:**

1. Results on four datasets are presented on multi-view tasks.
2. Empirical study shows that the method consistently improves the performance of downstream tasks for both few-shot and routine tasks.

**Weaknesses:**

1. Why two modulation processes are useful in this task was unclear.  Also, the benefit of the TOMRL on different dataset domain is also not discussed or analyzed in the paper. I think the paper should at least study at least one scenario , e.g., NoisyFashion to Caltech 101-7, to verify the effectiveness of TMORL as this is considered as one of the main contribution of the paper. It shall also be helpful to analyze why TOMRL is helpful in learning a high-quality unified representation, perhaps from the perspective of gradient analysis.

2. The display of experimental results in this paper is not uniform. For example, bold results in some tables indicate the best results, while others denote the results of TOMRL. Please unify the form in the full text. Alternatively, give clear comments in each table title.

**Questions:**

Regarding the cross task experiment in Table 3, the proposed TOMRL brings a decrease in NMI indicators of NoisyFashion to EdgeFashion. With the significant growth MORL has brought under other conditions, why did this particular decline occur?

---

> ### Author Response · Authors · 2023-11-23
> **Response to Reviewer V7F6**
>
> Thank you for your constructive comments.
>
> **Response to Weakness 1 (Different dataset domain):**
>
> Thank you for your valuable suggestion on ''experimenting on different dataset domains’’. The performance of TOMRL in cross-domain scenarios has been validated on two variant datasets, NoisyFashion and EdgeFashion, in Sec. 5.3 of the manuscript. The experiment setting of ``cross-domain between NoisyFashion (2 views) and Caltech101-7 (6 views)'', mentioned by the reviewer, is challenging for the current framework. This challenge comes from the fact that existing multi-view models are sensitive to the number of views. Although the gradient-based task embedding (in Sec. 3.1.2) provides flexibility in handling tasks with different scales (i.e., ways and shots), TOMRL, and in particular the meta-learner, is still based on the existing multi-view models. Therefore, the key to implementing the cross-domain "between NoisyFashion and Caltech101-7" is to design a unified model that can handle data with different numbers of views, which is not the focus of this work but is worth investigating and is part of our future work.
>
> **Response to Weakness 2:**
>
> Thank you for your valuable suggestions. The revised manuscript explains the special markings in each table title.
>
> **Response to Questions:**
>
> We rechecked the experimental results. One possible reason is that TOMRL improves the number of samples clustered correctly, but does not significantly improve the cluster distribution obtained by InfoDDC in this scenario.

---

### Official Review · Reviewer_caBF · 2023-11-05

**Soundness:** 2 fair
**Presentation:** 2 fair
**Contribution:** 2 fair
**Rating:** 3
**Confidence:** 4

**Summary:**

The paper proposes a multi-view representation learning method, where the key idea is to modulate features in the View-specific encoding and Fusion modules according to the task guidance. The authors design a gradient-based embedding strategy to represent multi-view tasks. In addition, a meta-learner is trained to map the task embedding into a set of view-specific parameters and a view-shared parameter. This whole process is formalized as a nested optimization problem and ultimately solved by a bi-level optimization scheme.

**Strengths:**

1. The paper proposes a task-oriented multi-view representation Learning method from a meta-learning perspective. The performance of classification and clustering tasks is improved significantly.
2. The proposed method defines an unsupervised multi-view task in an episode fashion, and designs a meta-learner for modulating the view-specific features and unified entity representations with the task guidance.
3. The proposed method models meta-learning and multi-view learning as a nested bi-level optimization.

**Weaknesses:**

1. For ‘representations from multiple views can better serve the task’ in the contribution 1. In fact, it explores the relationship amongst various tasks, from the perspective of multi-task learning, which is not enough as an innovation point.
2. In the section 3, the authors mentioned that the fusion process of features may also be inconsistent. The proposed method focuses on Fusion modules, how does the author align the features of similar instances in different tasks?
3. The authors should discuss the insight of this paper with the lifelong multi-view learning or multi-view multi-task learning.
4.  In term of the loss function, the author does not introduce the concept of weight. For multiple tasks, the data distribution and importance of different tasks are different. How does the author solve this problem?
5. The current manuscript need to be carefully polished, such as, in the section 2, 〖R〗^(d_H ) not R_(d_H ), in table 2, line 4, the font thickness should be consistent； the notations in equation (1)

**Questions:**

Please check the comments above.

---

> ### Author Response · Authors · 2023-11-23
> **Response to Reviewer caBF**
>
> Thank you for your constructive comments, which were inspirational in refining the manuscript.
>
> **Response to Weaknesses 1 & 3 (Connections and differences with multi-view multi-task learning (MVMTL) and continuous/lifelong multi-view learning (CMVL)):**
>
> MVMTL, CMVL, and our TOMRL all consider both multi-view and multi-task scenarios, exploiting comprehensive feature representation of multiple views in each task as well as the task relationships of multiple related tasks. However, they are different in terms of learning objectives, key challenges, and practical settings.
>
> - **MVMTL follows the multi-task learning setting**, where the learning objective is to improve performance on the main task through joint training on multiple related (auxiliary) tasks. Key challenges include selecting/designing auxiliary tasks, balancing loss functions across multiple tasks (Weakness 4), etc. The counterpart is conventional (single-view) multi-task learning, with the difference that MVMTL utilizes comprehensive information (e.g., consistency and complementarity, etc.) from multiple views of each task. Once the multi-view information is well integrated, most MVMTL methods degrade to the conventional multi-task scenario [1].
> - **CMVL follows the continuous learning setting**, where the learning objective is to handle new tasks without performance degradation on previously learned tasks for a series of consecutive tasks. The key challenge is to overcome catastrophic forgetting. Similarly, the counterpart is conventional (single-view) continuous learning, with the difference that CMVL utilizes comprehensive information from multiple views of each task.
> - **Our TOMRL follows the meta-learning setting**, where the learning objective is to rapidly handle new downstream tasks by learning meta-knowledge over multiple tasks. To this end, we construct task-level training and test sets on the multi-view dataset and train the model in an episodic fashion. The counterpart is conventional (single-view) meta-learning, with the difference that TOMRL considers both view-specific representations and fused unified representations in the meta-learning process. If the multi-view properties of the task are ignored, TOMRL degrades to a regular meta-learning approach.
>
> In addition, we review existing works involving ''meta-learning'' and ''multi-view learning'' in the manuscript, and most of them aim to improve the performance of meta-learning methods by utilizing multi-view information rather than the fast adaptation of multi-view representations in different downstream tasks.
>
> [1] Lu, Run-kun, et al. Multi-view representation learning in multi-task scene. *Neural Computing and Applications* 32 (2020): 10403-10422.
>
> **Response to Weakness 2 (Feature alignment of similar instances in different tasks):**
>
> We apologize for some misunderstandings due to unclear writing in the manuscript. Firstly, the basic principle ‘’b) the fusion process of features may be inconsistent" mentioned in Section 3 means that ‘’given multiple view-specific features, the unified representation fused for different downstream tasks should be different using the same fusion strategy’’, rather than ‘’the fusion strategy being performed in different tasks should be different’’. In contrast to the latter, the former does not suffer from feature misalignment due to the same fusion strategy. Existing multi-view representation learning methods pay little attention to this unified representation inconsistency, i.e., once the feature extraction and fusion strategies are given, the unified representation is determined. Part of our motivation comes from this gap between existing methods and the basic principle (the other part comes from the basic principle (a) in Section 3). Secondly, our TOMRL inherits model-agnostic advantages from gradient-based meta-learning, which makes it flexible enough to be integrated into a variety of multi-view learning models and, of course, a variety of fusion strategies.
>
> **Response to Weakness 4 (Weights of the loss function):**
>
> In the response to Weakness 1, we discussed the differences between the multi-view multi-task learning methods and the proposed work. The weighting of the loss function is a key issue in jointly training multiple tasks, but is not a concern in TOMRL, which follows a meta-learning "episode" training fashion.
>
> **Response to Weakness 5 (Notation mistakes):**
>
> Thank you very much for your constructive suggestions. These notation mistakes have been revised in the revised manuscript.

---

### Official Review · Reviewer_aCu2 · 2023-12-04

**Soundness:** 2 fair
**Presentation:** 2 fair
**Contribution:** 1 poor
**Rating:** 3
**Confidence:** 4

**Summary:**

This paper introduces a task-oriented multi-view representation learning method. Specifically, it adopts the meta-learning paradigm to minimize the distribution differences in representations across various tasks. However, the overall technical soundness of this paper appears to be lacking, and the experimental evidence presented is not sufficiently convincing.

**Strengths:**

1. For multi-view representation learning, different tasks have distinct requirements for the distribution of representations. This paper highlights this issue.

2. The proposed method integrates meta-learning and multi-view learning through a nested bi-level optimization approach.

**Weaknesses:**

1. The framework and methodology lack innovation, constituting a simple combination and minor extension of existing works without significant theoretical contributions.

2. The experimental section of the paper appears to be too simplistic and insufficient. The selected datasets are on the smaller side, limiting the ability to validate the model's effectiveness and scalability in more complex scenarios.

3. In the cross-domain experiments of this article, validation was conducted solely on the Noisy Fashion and Edge Fashion datasets. The domain difference between the two is minimal, almost negligible.

4. The paper only reports results under the 5-way 5-shot setting and lacks evaluation in more diverse few-shot scenarios, such as 1 shot or 10 shot. Therefore, it becomes challenging to assess the method's generalization capability across varying data quantities.

**Questions:**

1. Due to the limited sample size in the dataset, avoiding overfitting poses a central challenge in few-shot learning. However, the paper lacks an in-depth discussion on this issue and does not introduce method-level strategies to alleviate the problem.

2. The authors should consider validating their method on the dataset with a significantly larger domain difference.

3. The introduction of multi-view seems to be optional. Additionally, there is no information exchange between the multi-view sub-networks.

---

### Meta-Review · Area_Chair_ZEFt · 2023-12-05

**Metareview:**

The paper explores a meta-learning-based architecture for multi-view representation learning, yielding promising performance in both clustering and classification tasks.

Highlights of the paper mentioned by the reviewers include:
1.The paper proposes a task-oriented multi-view representation learning method from a meta-learning perspective.
2.The proposed method in this paper significantly enhances the experimental performance of both clustering and classification tasks simultaneously.

The main concerns expressed by the reviewers include:
1.The paper faces limitations in innovating model design.
2.The motivation behind this paper remains unclear, which hinders a comprehensive understanding of the work.
3.The experimental methodology is insufficient, notably due to a constrained sample size, a lack of diversity in datasets, and experiment settings that are not convincingly established.

In the rebuttal period, AC additionally invited two domain experts to reevaluate this submission. The paper finally received five scores (i.e., 3,5,3,6,3) from reviewers. After carefully reading the manuscript, comments, and the corresponding response, I agree with the reviewers that this paper is not novel enough to be accepted.

**Justification For Why Not Higher Score:**

The main concerns expressed by the reviewers include:
1.The paper faces limitations in innovating model design.
2.The motivation behind this paper remains unclear, which hinders a comprehensive understanding of the work.
3.The experimental methodology is insufficient, notably due to a constrained sample size, a lack of diversity in datasets, and experiment settings that are not convincingly established.

**Justification For Why Not Lower Score:**

Highlights of the paper mentioned by the reviewers include:
1.The paper proposes a task-oriented multi-view representation learning method from a meta-learning perspective.
2.The proposed method in this paper significantly enhances the experimental performance of both clustering and classification tasks simultaneously.

---

### Decision · Program_Chairs · 2024-01-16

Reject